# Anti-Obesity Effects and Changes of Fecal Microbiome by Lactic Acid Bacteria from Grains in a High-Fat Diet Mouse Model

**DOI:** 10.3390/ijms26189056

**Published:** 2025-09-17

**Authors:** Chang Woo Jeon, Hyeon Yeong Lee, Hong Sik Kim, Min Ju Seo, Kye Won Park, Jung-Hoon Yoon

**Affiliations:** Department of Food Science and Biotechnology, Sungkyunkwan University, 2066 Seobu-ro, Jangan-gu, Suwon 16419, Republic of Korea; novag303@skku.edu (C.W.J.); 627hyeon@naver.com (H.Y.L.); hongsiik29@icloud.com (H.S.K.); yeondoli@naver.com (M.J.S.); kwpark@skku.edu (K.W.P.)

**Keywords:** probiotics, *Latilactobacillus*, obesity, lipid metabolism, microbiome

## Abstract

Single three-lactic acid bacterial strains with anti-adipogenic effects in C3H10T1/2 cells and possessing beneficial probiotic properties were administered to mice fed a high-fat diet. Of the three strains, *Lactiplantibacillus plantarum* RP12, which had the lowest weight gain, was utilized for further studies, including a second mouse experiment lasting 10 weeks. Oral administration of *Lactiplantibacillus plantarum* RP12 resulted in reduced body weight gain and epididymal fat mass. Significant reductions in serum total cholesterol, triglycerides, and blood glucose were observed in the group treated with *Lactiplantibacillus plantarum* RP12. This strain was found to regulate the expression of genes associated with lipid metabolism in epididymal adipose tissue and liver. It induced changes in the composition of fecal microbiota. Although there is no difference in the *Bacillota* to *Bacteroidota* ratio between the HFD and RP12 groups, notable differences in the compositions at the family, genus, and species levels were evident. Specifically, differences in the proportions of some taxa reported to have an association with obesity were observed between the HFD and RP12 groups. Fecal analyses demonstrated that *Lactiplantibacillus plantarum* RP12 diminishes lipid absorption and augments the production of short-chain fatty acids in the intestine. *Lactiplantibacillus plantarum* RP12 also mitigated damage to the morphology of the ileum and colon caused by a high-fat diet and promoted the expression of *Claudin-1* and *Muc2*. Overall, *Lactiplantibacillus plantarum* RP12 has potential as a useful probiotic to address metabolic disorders as well as obesity, substantiating the positive in vivo indicators and modulation of gut microbiota in a high-fat diet-induced obese mouse model.

## 1. Introduction

Obesity is a multifaceted health issue influenced by abnormal metabolic processes, which is influenced by psychosocial, genetic, and environmental elements [1,2]. The dramatic increase in obesity over the decades poses a substantial threat to public health, contributing to metabolic syndrome, diabetes, fatty liver disease, cardiovascular diseases, and other conditions [3,4]. A range of methods, including dietary, pharmacological, and surgical interventions, have been employed to manage obesity, and new drugs have recently garnered considerable attention [5,6]. Many recent studies have robustly linked obesity with alterations in the gut microbiome, characterized by specific bacterial compositions and functional changes [7,8,9]. These obesity-induced changes in the gut’s microbial makeup are linked with enhanced dietary energy extraction [10] and modifications in fatty acid metabolism in adipose tissues and the liver [11,12]. Therefore, modulating the gut microbiota’s composition could offer a valuable approach to managing metabolic disorders and obesity [8,13,14].

Probiotics are live microorganisms that provide health advantages to the host when consumed in sufficient amounts and are recognized for their ability to enhance or restore the gut microbiota [15]. The health benefits include lowering blood cholesterol and hypertension, regulating the immune system, and managing intestinal inflammatory diseases [15,16]. Several studies have shown that probiotics may also have anti-adipogenic potential or prevent excessive lipid accumulation [12,17,18]. Probiotics are considered a promising alternative for the prevention of metabolic diseases, including obesity, and their mode of action potentially involves the modulation of the gut microbiota composition, thereby leading to positive changes in the intestinal environment [15,19]. The abilities to inhibit fatty acid synthesis, reduce inflammation in vivo, and produce short-chain fatty acids are also crucial roles of probiotics in preventing or treating metabolic diseases [12,14,20]. Nevertheless, there remains a need for the development of new probiotic strains that offer enhanced benefits for improving gut health and addressing dysbiosis in gut microbiota.

Grains are rich in fiber and oligosaccharides, making them outstanding sources of prebiotics [21]. In Asia, grains represent the most widely consumed staple food, highlighting their significance as a prebiotic source for the population. Lactic acid bacteria (LAB) inhabiting grains may be particularly effective at utilizing the carbohydrates present as prebiotics, and several LAB species have been identified within them [21,22]. Nevertheless, limited research has been conducted on the anti-obesity effects of LAB isolated from grains.

In our recent study, LAB strains were isolated from four types of grains, and certain strains demonstrated enhanced anti-adipogenic effects in C3H10T1/2 cells, alongside fundamental probiotic properties such as viability in acid and bile salts and strong adhesion to Caco-2 cells [22]. This study aims to explore the anti-obesity effects of three LAB strains using various biomarkers in a high-fat diet mouse model. From the findings, *Lactiplantibacillus plantarum* RP12 has been confirmed as a potential probiotic agent due to its ability to inhibit adipogenesis, modulate the gut microbiome, and other beneficial properties.

## 2. Results

### 2.1. Effects of Lacticaseibacillus rhamnosus GG, Pediococcus pentosaceus K28, Lactiplantibacillus plantarum RP12, and Levilactobacillus brevis RP21 on Body

Over a period of 7 weeks, variations in body weight were monitored across different groups of mice. The group fed an HFD exhibited a more rapid weight gain (Appendix A). After 7 weeks, the average weight gain in the HFD group was 14.21 ± 1.83 g, whereas those in the LGG, K28, RP12, and RP21 groups were 12.70 ± 1.64 g, 11.88 ± 3.01 g, 10.43 ± 0.73 g, and 11.97 ± 1.75 g, respectively (Appendix A). Relative to the HFD group, the weight gains in the LGG, K28, RP12, and RP21 groups decreased by 10.6, 16.4, 26.6, and 15.8%, respectively (Appendix A). No significant differences in FER were observed between the LGG, K28. RP12 and RP21 groups compared to the HFD group. After 60 min, blood glucose levels were significantly lower in the RP12 and RP21 groups compared to the HFD group (*p* < 0.001 or 0.01) (Appendix A). The areas under the blood glucose response curve (AUC) for the LGG, RP12, and RP21 groups were significantly lower than that of the HFD group (*p* < 0.05 or 0.001) (Appendix A).

### 2.2. Effects of Lactiplantibacillus plantarum RP12 on Body, Visceral Organs, and Fat Tissue Weight 

In another experiment lasting 10 weeks and involving only HFD and RP12 groups, body weight changes were observed. After 10 weeks, the average weights of the HFD and RP12 groups were 36.59 ± 0.76 g and 31.57 ± 0.52 g, respectively (Figure 1A). Starting from week 4, substantial weight differences between the HFD and RP12 groups were noted, with significant levels noted (*p* < 0.05, 0.01, or 0.001) (Figure 1A). The average weight gain after 10 weeks in the RP12 group was significantly less, at 12.64 ± 0.53 g compared to 17.54 ± 0.55 g in the HFD group (*p* < 0.001) (Figure 1B). No significant difference was observed in food intake between the two groups (Figure 1C). The FER significantly decreased in the RP12 group as compared to the HFD group (Figure 1D). After 60 min, blood glucose levels in the RP12 group were significantly lower compared to the HFD group (*p* < 0.05), and the area under the blood glucose response curve (AUC) for the RP12 group was significantly lower than that of the HFD group (*p* < 0.01) (Figure 1E,F). No significant changes were observed in the weights of organs, including liver, spleen, and kidney, between the HFD and RP12 groups (Figure 1G). However, the epididymal fat mass significantly decreased in the RP12 group compared to the HFD group (*p* < 0.01) (Figure 1G).

### 2.3. Effects of Lactiplantibacillus plantarum RP12 on Serum Biochemical Parameters

The serum concentrations of total cholesterol and glucose in the RP12 group were significantly lower (*p* < 0.05) than those in the HFD group (Table 1). Compared with the HFD group, serum levels of TG, LDL, AST, and ALT were reduced in the RP12 group (Table 1; Appendix A). Total cholesterol decreased by 7.9% (*p* < 0.05) and glucose decreased by 35.9% (*p* < 0.05) in the RP12 group. Relative to the HFD group, there was a 3% increase in HDL and reductions of 11.4% in TG and 17.2% in LDL in the RP12 group, although these differences were not statistically significant. Both ALT and AST levels decreased in the RP12 group; however, the differences were not significant between the two groups (Appendix A).

### 2.4. Effects of Lactiplantibacillus plantarum RP12 on Genes Involved in Lipid Metabolism in Epididymal Fat Tissue and Liver

The results for the mRNA expression of genes involved in lipid metabolism in epididymal fat tissue and liver are presented in Figure 2. In the epididymal fat tissue of the RP12 group, there was a decrease in expression of certain genes associated with lipid metabolism compared to the HFD group (Figure 2A). When compared to the HFD group, the administration of *Lactiplantibacillus plantarum* RP12 significantly downregulated (*p* < 0.01 or 0.05) fatty acid synthetase (*Fas*), stearoyl-Coenzyme A desaturase 1 (*Scd1*), and CCAAT-enhancer-binding protein-*α* (*Cebpα*), which are genes related to fatty acid synthesis (Figure 2A). In the RP12 group, there was a significant decrease (*p* < 0.01 or 0.05) in the expression of genes for lipoprotein-lipase (*Lpl*) and cluster of differentiation 36 (*Cd36*), which are related to membrane transport (Figure 2A). Additionally, expressions of adipocyte protein 2 (*aP2*) and sterol regulatory element-binding protein-1C (*Srebp-1C*) were also reduced in the RP12 group compared to the HFD group (Figure 2A). The expressions of tumor necrosis factor-alpha (*Tnfα*) (*p* < 0.01), monocyte chemoattractant protein-1 (*Mcp1*) (*p* < 0.05), and interleukin 6 (*Il-6*) (*p* < 0.05), associated with pro-inflammatory cytokines, were significantly reduced in the RP12 group compared to the HFD group (Figure 2B). To explore the effects of *Lactiplantibacillus plantarum* RP12 on lipid metabolism in the liver, a comprehensive analysis of related gene expressions was conducted (Figure 2C). In comparison with HFD, the expressions of genes associated with lipid production, *Srebp1C* (*p* < 0.01), *Fas* (*p* < 0.01), and *Scd1* (*p* < 0.05), were notably decreased in the RP12 group (Figure 2C). In addition, there was a significant increase (*p* < 0.05) in the expressions of β-oxidation-related genes, including carnitine palmitoyltransferase1 alpha (*Cpt1α*), uncoupling protein (*Ucp2*), acyl coenzyme A oxidase 1 (*Aox1*), and acyl coenzyme A thioesterase 1 (*Acot1*), in the RP12 group (Figure 2D).

### 2.5. Effects of Lactiplantibacillus plantarum RP12 on Changes in Ratio and Composition of Fecal Microbiota

The alpha diversity indices (Shannon and Chao1) and beta diversity index (PCoA) between HFD and RP12 groups were calculated. No significant changes in alpha diversity were observed between the two groups (Appendix A). PCoA analysis using generalized UniFrac distances clearly demonstrated a statistically significant separation between the microbial communities of the HFD and RP12 groups (Appendix A).

Fecal compositions of the two dominant phyla, *Bacillota* and *Bacteroidota,* from the HFD and RP12 groups were compared after 10 weeks. The ratio of *Bacillota* and *Bacteroidota* was similar in both groups (Appendix A). No significant differences were observed in the relative abundances of *Bacillota* and *Bacteroidota* between the HFD and RP12 groups (Figure 3A). However, significant differences were found in the compositions of families, genera, and species between the two groups. Specifically, significant reductions (*p* < 0.05) were observed in the levels of *Streptococcaceae* and *Peptostreptococcaceae* in the RP12 group, whereas levels of *Coriobacteriaceae* (*p* < 0.09) and *Lactobacillaceae* (*p* < 0.01) increased in the RP12 group, as compared to the HFD group (Figure 3B). Significant FDR values (<0.05) were also observed in *Streptococcaceae*, *Peptostreptococcaceae*, and *Lactobacillaceae*.

LEfSe analysis was performed to identify specific bacterial genera and species that were dominant in the HFD and RP12 groups (Appendix A). Consequently, significant differences in the compositions of major genera and species were observed between the two groups (Appendix A). Specifically, the proportions of genera *Anaerotruncus*, *Romboutsia*, *Lactococcus*, *Harryflintia*, *Enterorhabdus*, and *Gemella* were significantly higher in the HFD group than in the RP12 group (Figure 4). Conversely, the proportions of genera *Lactobacillus*, *Olsenella*, and *Clostridium* were significantly higher in the RP12 group than in the HFD group (Figure 4; Appendix A). Significant FDR values (<0.05) were observed in the other genera except for *Harryflintia* and *Olsenella*. Significant decreases (*p* < 0.01 or 0.05) in levels of *Streptococcus acidominimus* and *Romboutsia timonensis* and significant increases (*p* < 0.01 or 0.05) in levels of *Olsenella* PAC001059_s, *Clostridium cocleatum*, *Bacteroides faecichinchillae*, *Lactobacillus brevis*, and *Lactiplantibacillus plantarum* were observed in the RP12 group (Figure 5). The changes of the six species, except for *Olsenella* PAC001059_s, were supported by significant FDR values (<0.05). The elevated abundance of *Lactiplantibacillus plantarum* in the RP12 group might be attributed to the administration of *Lactiplantibacillus plantarum* RP12 followed by its colonization in the intestine.

### 2.6. Concentrations of Lipids and Short-Chain Fatty Acids in Feces

The lipids and three short-chain fatty acids (acetic acid, propionic acid, and butyric acid) were quantitatively analyzed in the feces collected from mice after 10 weeks. The fecal lipid content (21.74 ± 0.26 mg/g) in the RP12 group was 41.5% higher than that (15.36 ± 0.30 mg/g) in the HFD group (Figure 6). The RP12 group demonstrated only a minor increase in acetic acid levels compared to the HFD group (Figure 6). Levels of butyric acid and propionic acid were significantly higher (*p* < 0.05) in the RP12 group than in the HFD group (Figure 6).

### 2.7. Histological Assessment of Colon and Ileum

The effects of administering *Lactiplantibacillus plantarum* RP12 on the intestine were evaluated by hematoxylin and eosin staining of the colon and the ileum, with histological parameters displayed in Appendix A. The findings indicated that the villi in both the ileum and colon of the HFD group were damaged and shortened, whereas in the RP12 group, the villi morphology in these structures was well preserved. The mRNA expression levels of two genes, *Claudin-1* and *Muc2*, in the colon and ileum are depicted in Figure 7A,B. In comparison to the HFD group, the group treated with *Lactiplantibacillus plantarum* RP12 exhibited significant increases (*p* < 0.05 or *p* = 0.05–0.07) in the expression of *Claudin-1* and *Muc2* (Figure 7A,B).

## 3. Discussion

Probiotics have been garnering attention as alternatives to pharmacological drugs, which may lead to serious side effects in the treatment of obesity [14,23,24]. The metabolic alleviation of obese phenotypes by probiotics has been shown to be induced mainly through host metabolism and gut microbiota modulation [12,25,26]. Lactic acid bacteria with anti-obesity effects have been isolated from various habitats, including intestines, kimchi, breast milk, and fermented foods [24,27,28,29]. In our previous study, numerous lactic acid bacteria were isolated from grains and subjected to an anti-adipogenic assay using C3H10T1/2 cells [22]. Of these, three lactic acid bacterial strains, *Pediococcus pentosaceus* K28, *Lactiplantibacillus plantarum* RP12, and *Levilactobacillus brevis* RP21, were identified as potential probiotic candidates due to their useful probiotic properties and anti-adipogenic effects in in vitro assays. The current study evaluated the anti-obesity effects of these three LAB strains in an obese mouse model. In mice experiments involving the three LAB strains, *Pediococcus pentosaceus* K28, *Lactiplantibacillus plantarum* RP12, and *Levilactobacillus brevis* RP21, there was a significant decrease in weight gain compared with the LGG and HFD group (Appendix A). Specifically, *Lactiplantibacillus plantarum* RP12 exhibited a more pronounced anti-obesity effect (showing 14–15% differences in weight gain) and significant improvement in glucose tolerance and AUC compared with the other two strains (Appendix A). These observations led strain RP12 to further investigations, including a second mouse experiment for 10 weeks. After 10 weeks, the average weight gain of the RP12 group decreased significantly (*p* < 0.001) by 27.9% compared with the HFD group (Figure 1B). *Lactiplantibacillus plantarum* RP12 significantly (*p* < 0.01) reduced epididymal fat mass and slightly reduced the liver weight (Figure 1G). These results confirm that *Lactiplantibacillus plantarum* RP12 exhibits significant anti-obesity effects. Several studies have demonstrated that lactic acid bacteria influence lipid metabolism in adipose tissue by regulating the expression of lipid metabolism-related enzymes [12,30,31]. A reduction in gene expressions related to lipid metabolism in epididymal adipose tissue and a decrease in pro-inflammatory gene expression in the same tissue were also observed in this study. These results corroborate previous findings (Figure 2A,B). Although no significant weight change was observed in the liver between the HFD and RP12 groups, significant decreases in the expression of genes related to lipid production and significant increases in the expression of β-oxidation-related genes were observed in the liver of the RP12 group (Figure 2C,D). These outcomes in the liver, treated with LAB, have also been documented previously [12]. Accordingly, *Lactiplantibacillus plantarum* RP12 is anticipated to exhibit anti-obesity effects by reducing lipid accumulation in adipose tissue and alleviating chronic hypo-inflammation in the same tissue (Figure 2). *Lactiplantibacillus plantarum* RP12 decreased levels of total blood cholesterol, LDL, and blood glucose. Elevated levels of total blood cholesterol and blood glucose have consistently been linked to obesity and dyslipidemia in the obesity-induced model [32]. Several LAB strains have been discovered to lower blood cholesterol levels [33,34,35]. It has been demonstrated that cholesterol not absorbed in the small intestine may be transformed into other compounds by gut microbes, thereby reducing cholesterol levels in the body [36]. In this study, it is proposed that the gut microbiota, modulated by *Lactiplantibacillus plantarum* RP12, may contribute to lower blood cholesterol levels.

Changes in the gut microbiota profile may control obesity by affecting energy harvesting and storage [24]. Many studies have confirmed changes in the diversity and composition of gut microbiota in obesity states [37,38]. In this study, no significant differences in diversity indices and richness estimators of the fecal microbiota were found between the RP12 and HFD groups, although clear differences in the composition of the dominant microbial community in the feces were observed between the two groups (Figure 3, Figure 4 and Figure 5). After long-term administration of a high-fat diet, dysbiosis of the gut microbiota was observed, contrasting with the RP12 group (Figure 3, Figure 4 and Figure 5). Although there was no clear difference in the proportions of the phyla *Bacillota* and *Bacteroidota* between the RP12 and HFD groups, the two groups exhibited distinct differences in the relative abundances of certain taxa at lower taxonomic levels. At more detailed taxonomic levels, both the HFD and RP12 groups showed alterations in several bacterial taxa associated with obesity-related dysbiosis (Figure 3, Figure 4 and Figure 5). The proportions of two families, *Streptococcaceae* and *Peptostreptococcaceae*, were shown to increase in mice fed a high-fat diet [39,40]. Specifically, the increase in the family *Streptococcaceae* is known to be associated with the development of obesity, metabolic disorders, and diabetes [41,42]. The proportion of the genus *Lactococcus*, belonging to the family *Streptococcaceae*, is significantly correlated with inflammation and insulin resistance [42]. A reduction in *Streptococcaceae*/*Lactococcus* may play a crucial role in preventing metabolic disorders [42]. The family *Coriobacteriaceae* is reported to produce short-chain fatty acids, particularly butyric acid [43,44,45], and is considered a potential contributor to various beneficial functions, such as glucose homeostasis and bile acid and lipid metabolism, in the host [46]. Accordingly, differences in the abundance of several families between the HFD and RP12 groups might influence metabolic properties, consistent with previous findings. The differences in the composition of major bacterial genera and species between the HFD and RP12 groups were evident from LEfSe analysis (Appendix A). The genus *Anaerotruncus*, which is prevalent in the HFD group, is known to be associated with obesity [47,48]. It is significantly positively correlated with liver weight gain and the accumulation of epididymal or perirenal fat, and significantly negatively correlated with fecal SCFA levels [49]. The genus *Romboutsia* was found to be significantly increased in the obese group and was positively associated with blood glucose levels, fat intake ratios, and BMI [50,51]. A *Romboutsia*-enriched microbiota displayed dysbiosis-like features, unlike the commensal group [50]. Additionally, *Enterorhabdus* and *Gemella* have also been reported to be positively associated with the prevalence of obesity [52,53]. The genus *Olsenella*, abundant in the RP12 group, is known to be diminished in patients with inflammatory bowel disease (IBD) and in high-fat-diet-induced groups, and serves as beneficial bacteria for SCFA production [54,55]. *Clostridium cocleatum*, also abundant in the RP12 group, showed a significant increase following metformin treatment in HFD mice and was positively correlated with several metabolic biomarkers [56]. *Bacteroides faecichinchillae* is commonly found in non-obese individuals compared to obese individuals and has a significant association with a lean body type [27,57]. The findings from this study demonstrate that changes in specific gut microbiota are correlated with biomarker outcomes in our mouse model, confirming the mechanism of obesity inhibition.

In this study, the fecal lipid content of the RP12 group was analyzed to be higher than that of the HFD group (Figure 6A). Variations in the absorption of lipids in the intestines may serve as a possible explanation for the reduced weight gain [58]. It is possible that *Lactiplantibacillus plantarum* RP12 can suppress obesity induced by a high-fat diet by inhibiting lipid absorption. Short-chain fatty acids (SCFAs) have been shown to prevent body weight gain induced by a high-fat diet through the modulation of gut microbiota and their beneficial roles in host health [59,60]. Changes were observed in the concentrations of SCFAs such as acetic acid, propionic acid, and butyric acid in feces between the RP12 and HFD groups (Figure 6B). Certain gut microbes are capable of producing SCFAs through the digestion of various types of carbohydrates [61,62]. It has been reported that *Lactobacillus* species can indirectly enhance the production of SCFAs through modulation of the gut microbiota, as demonstrated in this study [27,63]. Research has indicated that butyric acid and propionic acid reduce food intake and obesity induced by a high-fat diet, and help prevent glucose intolerance [64,65,66]. Our findings suggest that the modulation of several taxa in the gut by *Lactiplantibacillus plantarum* RP12 may influence the production of SCFAs as well as changes in microbial taxa. An increased proportion of *Clostridia*, an important producer of butyric acid in the intestine [27,67], was observed in the RP12 group, suggesting that *Lactiplantibacillus plantarum* RP12 plays a role in this increase of *Clostridia*. It has also been shown that the intake of *Lactiplantibacillus plantarum* RP12 could mitigate damage to the intestinal barrier caused by a high-fat diet (Appendix A), even though no quantitative morphometric analysis (e.g., villus height and crypt depth) or correlation with gene expression was performed. A high-fat diet may influence intestinal permeability by affecting bacterial overgrowth in the small intestine and impacting nervous and metabolic processes [68]. Gene expression of *Claudin-1* and *Muc2* in the colon and ileum tended to increase in the RP12 group compared to the HFD group (Figure 7). *Claudin-1* is one of the tight junction proteins that regulate permeability in the intestine, and its expression has been associated with a reduction in colon cancer [69]. *Muc2* is a mucin secreted from the ileum and colon, and its deficiency is associated with disruption of epithelial homeostasis and the development of colon cancer [70,71]. The mRNA expression of *Claudin-1* and *Muc2* was measured as a preliminary indication of barrier-related responses. However, because the focus of the work was on anti-obesity effects, protein-level confirmation and functional permeability assays were not performed and will be addressed in future studies. *Lactiplantibacillus plantarum* RP12, which was isolated from grains, may be particularly advantageous at utilizing the carbohydrates present as prebiotics in the gut. Nevertheless, its effectiveness appears to have limitation that requires tests on humans or animals in the future. Based on the findings of this study, *Lactiplantibacillus plantarum* RP12 is concluded to mitigate obesity by lowering the metabolic disturbances through alterations in biomarkers in the obese mouse model and modulation of several taxa in the gut.

## 4. Materials and Methods

### 4.1. Bacterial Strains and Growth Conditions

*Pediococcus pentosaceus* K28, *Lactiplantibacillus plantarum* RP12, *Levilactobacillus brevis* RP21, and *Lacticaseibacillus rhamnosus* GG were obtained from our previous study [22] and routinely cultivated for 24 h at 30 °C on De Man–Rogosa–Sharpe (MRS, BD Difco, Sparks, MD, USA) agar. Their cell mass was suspended in 20% (*v*/*v*) glycerol (Georgiachem, Norcross, GA, USA) and stored at −80 °C for long-term preservation.

### 4.2. Animals and Experimental Design

The animal care and studies of the mice were conducted in accordance with the guidelines and approval of the Institutional Animal Care and Use Committee (IACUC) of the College of Biotechnology at Sungkyunkwan University (approval date: 7 September 2019, approval number: SKKUIACUC-20-02-10-2), covering the entire study period (November 2020–September 2021) and reported in accordance with ARRIVE guidelines. For the pilot experiment, male C57BL/6J mice aged 5 weeks, which were procured from RaonBio Inc. (Yongin, Republic of Korea), were housed under controlled conditions (24 ± 2 °C temperature, 50 ± 10% humidity) with a 12 h light/dark cycle. Following a 1-week acclimation period, 5-week-old mice were randomly assigned to 5 groups (n = 3/group): high-fat diet (HFD), high-fat diet plus *Lacticaseibacillus rhamnosus* GG (LGG), high-fat diet plus *Pediococcus pentosaceus* K28 (K28), high-fat diet plus *Lactiplantibacillus plantarum* RP12 (RP12), and high-fat diet plus *Levilactobacillus brevis* RP21 (RP21). The HFD group received a high-fat diet (HFD, 60% of energy from fat, 21.9 kJ, RaonBio Inc.) for 7 weeks. Concurrently, the LGG, K28, RP12, and RP21 groups received the same HFD for 7 weeks and received daily oral doses of *Lacticaseibacillus rhamnosus* GG, *Pediococcus pentosaceus* K28, *Lactiplantibacillus plantarum* RP12, and *Levilactobacillus brevis* RP21, respectively.

In a subsequent experiment using *Lactiplantibacillus plantarum* RP12, male C57BL/6J mice aged 5 weeks were purchased from RaonBio Inc. (Yongin, Republic of Korea) and maintained in controlled conditions (24 ± 2 °C temperature, 50 ± 10% humidity) with a 12-h light/dark cycle. After a 1-week acclimation period, these 5-week-old mice were randomly divided into 2 groups (n = 7/group): high-fat diet (HFD) and high-fat diet plus *Lactiplantibacillus plantarum* RP12 (RP12). The number of mice was determined based on our previous study [27] with a similar experimental design to ensure comparability and adequate statistical power. The HFD group received a high-fat diet (HFD, 60% of energy from fat, 21.9 kJ, RaonBio Inc.) for 10 weeks. The RP12 group, likewise on an HFD for 10 weeks, received *Lactiplantibacillus plantarum* RP12 through daily oral administration.

Live LAB cells were administered daily via oral gavage at a concentration of 10^9^ CFU per 200 μL 0.85% saline, as recommended by the WHO and the Korea Food and Drug Administration. Throughout the experiment, food intake and body weight were monitored weekly. Fecal samples were collected after 10 weeks and stored at −80 °C. The food efficiency ratio (FER) was calculated as the total body weight gain from the diet divided by the total diet consumed during the animal experiments. For the glucose tolerance test (GTT), mice were fasted for 12 h in the 9th week. Blood glucose levels were assessed from tail vein blood at intervals of 0, 15, 30, 60, 90, 120, 150, and 180 min following intraperitoneal glucose injection (2 g/Kg). At the conclusion of the experiment, the mice were fasted for 16 h and euthanized under anesthesia by exposure to carbon dioxide (CO_2_). Following euthanasia, the visceral organs (liver, spleen, kidney, colon, and ileum) and the epididymal fat pad were collected and weighed. The epididymal fat pad, liver, colon, and ileum were preserved by freezing in liquid nitrogen for subsequent genetic analysis. Blood was drawn via cardiac puncture and centrifuged for 10 min at 3000 rpm to separate serum.

### 4.3. Serum Analysis

Levels of alanine transaminase (ALT), aspartate transaminase (AST), total cholesterol, glucose, triglyceride (TG), high-density lipoprotein (HDL), and low-density lipoprotein (LDL) were measured using a biochemical automatic analyzer (AU480, Beckman Coulter Inc., Brea, CA, USA) following the manufacturer’s instructions.

### 4.4. RNA Extraction and Quantitative Real-Time Polymerase Chain Reaction (RT-PCR)

Total RNA extraction from epididymal fat tissue, liver, ileum and colon was performed using an RNeasy Mini Kit (Qiagen, Hilden, Germany) and TRIzol (Invitrogen, Carlsbad, CA, USA) according to the manufacturer’s protocol. First-strand complementary DNA was synthesized using a Veriti™ 96-Well Thermal Cycler machine (Thermo Scientific, Waltham, MA, USA) by mixing the total RNA with ReverTra Ace Master Mix (Toyobo, Osaka, Japan). A mixture of Power SYBR Premix ExTaq (RP041A; Takara, Shiga, Japan), primers, and cDNA was employed for amplification using a thermal cycler machine (Takara). Normalization of gene expression was performed using a housekeeping gene, *36B4*. The primer sequences for eighteen genes used in this study are shown in previous studies [12,72,73,74,75,76,77] or Appendix A. The primer sequences for other genes are detailed in an earlier study [12].

### 4.5. Analysis of 16S rRNA Gene Sequences from Gut Microbiome

For microbiome analysis, genomic DNA was extracted from fecal samples using a QIAamp DNA Stool Mini Kit (Qiagen, Hilden, Germany) following the manufacturer’s protocol. The initial and second amplifications were conducted as previously reported [12]. The sequencing was carried out according to the method of Chunlab Inc. (Seoul, Republic of Korea) using an MiSeq sequencing system (Illumina, San Diego, CA, USA). Taxonomic profiling and sequencing data analysis were performed using the Illumina platform (Chunlab Inc.) and as previously described [27]. Alpha diversity was assessed using OTU information and is expressed via the Chao 1 and Shannon index. The structure of the microbiota across different groups was analyzed using principal coordinate analysis (PCoA) at the genus level, utilizing the beta diversity index. Differences between the two groups were tested using PERMANOVA with Euclidean distances and 999 permutations [78]. The linear discriminant analysis effect size (LEfSe) technique was implemented using a Latent Dirichlet Allocation (LDA) score threshold of 3.0 and a *p*-value < 0.05. The relative abundance (%) of bacteria at various taxonomic levels was quantified and compared.

### 4.6. Quantitative Analyses of Lipids and Short-Chain Fatty Acids

Lipids in feces (about 0.4 g) taken from mice were extracted and analyzed as previously outlined in reference [79]. Measurement of short-chain fatty acids (SCFAs) was conducted as previously described [27].

### 4.7. Histological Analysis of the Colon and Ileum

Tissues from the colon and ileum were stained with hematoxylin and eosin (H&E) according to methods described earlier in reference [80].

### 4.8. Statistical Analysis

Statistical analyses were performed using SPSS version 19.0 (SPSS Inc., Chicago, IL, USA). Data are presented as mean ± SEM. Statistical significance in gene expression differences between experimental groups in animals was determined by an unpaired Student’s t-test. For relative abundance analysis of the gut microbiome, significant differences between groups were assessed using the Wilcoxon rank-sum test. Values were considered statistically significant when *p* < 0.05.

## 5. Conclusions

In this study, three lactic acid bacterial strains were used to investigate the anti-obesity effect in a mouse model subjected to a high-fat diet, and among these, *Lactiplantibacillus plantarum* RP12 was chosen for the various studies, including an additional mouse experiment. *Lactiplantibacillus plantarum* RP12 was found to inhibit adipogenesis by regulating the gene expressions related to lipid metabolism in the epididymal adipose tissue and liver of high-fat diet mice. Fecal analyses indicated that *Lactiplantibacillus plantarum* RP12 could exert its anti-obesity effects through the modulation of several taxa in the gut and by reducing lipid absorption and enhancing SCFA production in the intestine. It may be a viable alternative for alleviating metabolic disorders and obesity caused by dysbiosis. Further research and clinical trials are necessary to assess its applicability and efficacy in humans.

## Figures and Tables

**Figure 1 ijms-26-09056-f001:**
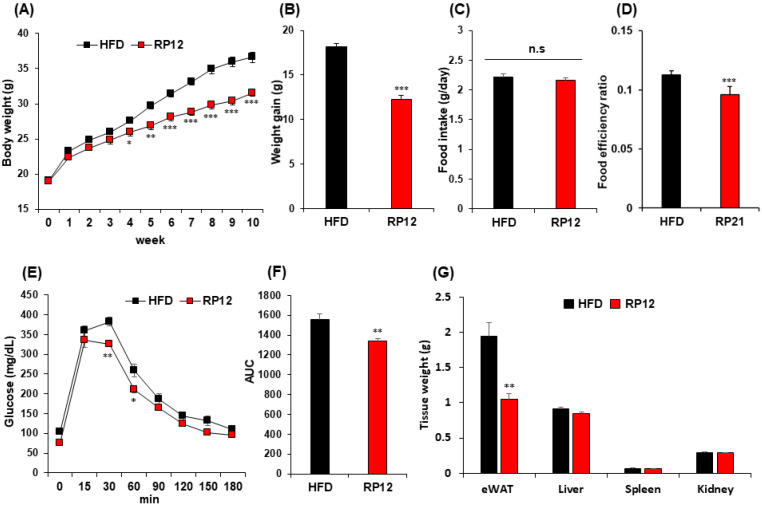
Effect of administered *Lactiplantibacillus plantarum* RP12 on high-fat diet-induced obese mouse model: (**A**) Weight change of mouse groups over 10 weeks. (**B**) Total weight gain for each group after 10 weeks. (**C**) Food intake for each group over 10 weeks. (**D**) Food efficiency ratio over 10 weeks for all groups. (**E**) Glucose tolerance test. (**F**) Area under curve. (**G**) Organ weights of mice in two groups after sacrifice. eWAT, epididymal white adipose tissue. Mice were fasted for 12 h before intraperitoneal glucose injection (2 g/kg). Results are presented as mean ± SEM (*n* = 7 per group). Significant differences between HFD and RP12 groups are indicated as * *p* < 0.05, ** *p* < 0.01, *** *p* < 0.001. No significant difference between HFD and RP12 groups is indicated as n.s.

**Figure 2 ijms-26-09056-f002:**
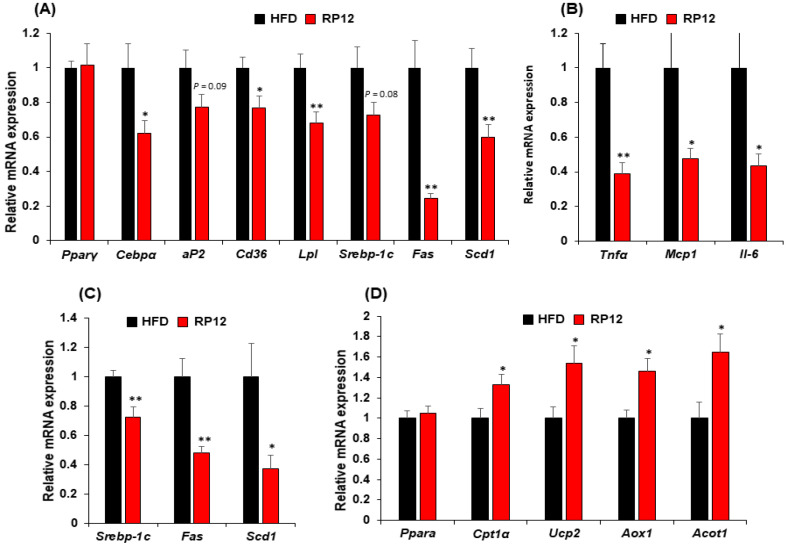
Effect of administered *Lactiplantibacillus plantarum* RP12 on gene expression in the epididymal fat pads and liver: (**A**) The mRNA expression levels of *Pparγ, Cebpα, aP2, Cd36, Lpl, Srebp-1c, Fas*, and *Scd1* in epididymal fat pads measured using quantitative real-time PCR. (**B**) Measurement of pro-inflammatory gene expression in epididymal fat pads via quantitative real-time PCR. (**C**) The mRNA expression levels of *Srebp-1c, Fas*, and *Scd1* in the liver determined by quantitative real-time PCR. (**D**) Assessment of fatty acid oxidation gene expression in liver via quantitative real-time PCR. *Ppar γ*, peroxisome proliferator-activated receptor γ; *Cebpα*, CCAAT-enhancer-binding protein-α; *aP2*, adipocyte protein 2; *Cd36*, cluster of differentiation 36; *Lpl*, lipoprotein lipase; *Srebp-1c*, sterol regulatory element-binding protein 1; *Fas*, fatty acid synthase; *Scd1*, stearoyl-CoA desaturase-1; *Tnfα*, tumor necrosis factor alpha; *Mcp1*, monocyte chemotactic protein 1; *Il-6*, interleukin-6; *Pparα*, peroxisome proliferator-activated receptor α; *Cpt1α*, carnitine palmitoyltransferase1 α; *Ucp2*, uncoupling protein 2; *Aox1*, acyl coenzyme A oxidase 1; *Acot1*, acyl coenzyme A thioesterase 1. Results are shown as mean ± SEM (*n* = 7 per group). Significant differences between HFD and RP12 are indicated as * *p* < 0.05, ** *p* < 0.01.

**Figure 3 ijms-26-09056-f003:**
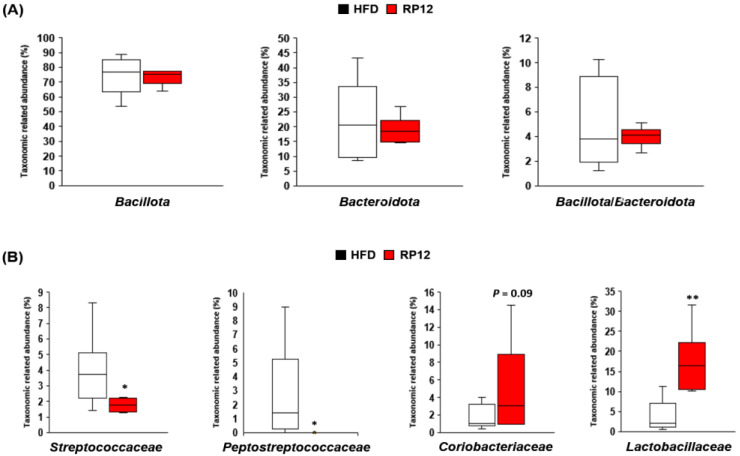
Effect of administered *Lactiplantibacillus plantarum* RP12 on fecal microbiome composition: (**A**) *Bacillota*, *Bacteroidota*, and *Bacillota* to *Bacteroidota* ratio at 10 weeks. (**B**) The relative abundance of specific families in fecal microbiota at 10 weeks. Results are shown as mean ± SEM (*n* = 7 per group). The nonparametric Wilcoxon signed-rank test for paired data and the Mann–Whitney U test for unpaired data were used. Significant differences between HFD and RP12 are indicated as * *p* < 0.05, ** *p* < 0.01.

**Figure 4 ijms-26-09056-f004:**
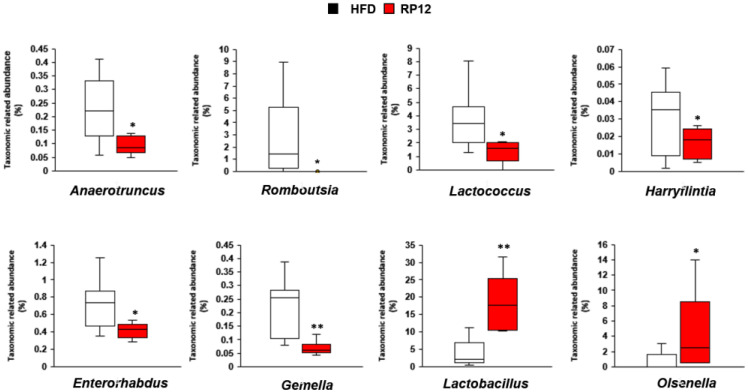
The relative abundance of specific bacterial genera in fecal microbiota at 10 weeks. Results are shown as mean ± SEM (*n* = 7 per group). The nonparametric Wilcoxon signed-rank test and Mann–Whitney U test were applied for paired and unpaired data, respectively. Significant differences between HFD and RP12 are indicated as * *p* < 0.05, ** *p* < 0.01.

**Figure 5 ijms-26-09056-f005:**
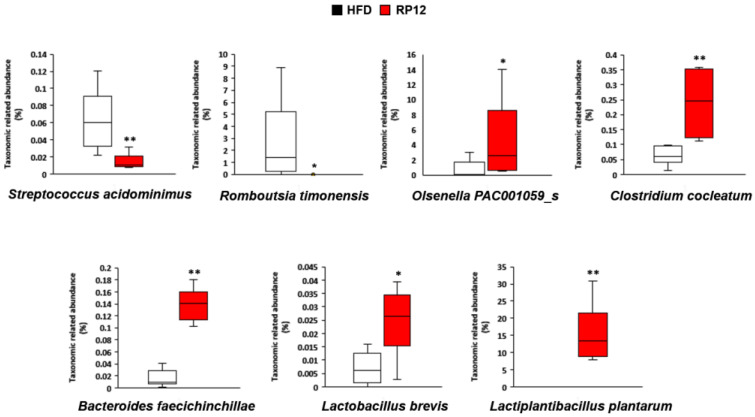
The relative abundance of specific bacterial species in fecal microbiota at 10 weeks. Results are shown as mean ± SEM (*n* = 7 per group). The nonparametric Wilcoxon signed-rank test for paired data and Mann–Whitney U test for unpaired data were used. Significant differences between HFD and RP12 are indicated as * *p* < 0.05, ** *p* < 0.01.

**Figure 6 ijms-26-09056-f006:**
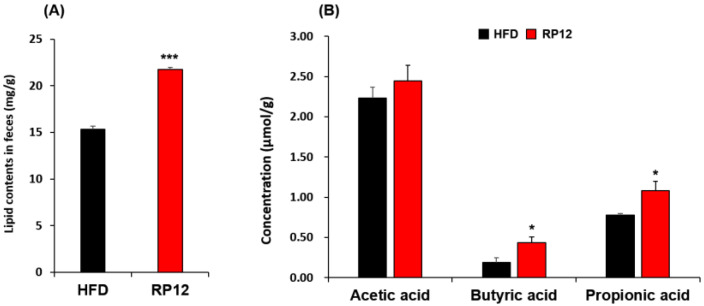
Concentrations of lipids and short-chain fatty acids (SCFAs) from fecal contents of HFD and RP12 groups: (**A**) Lipid concentration. (**B**) Concentration of acetic acid, butyric acid, and propionic acid. Results are presented as mean ± SEM (*n* = 7 per group). Significant differences between HFD and RP12 are indicated as * *p* < 0.05, *** *p* < 0.001.

**Figure 7 ijms-26-09056-f007:**
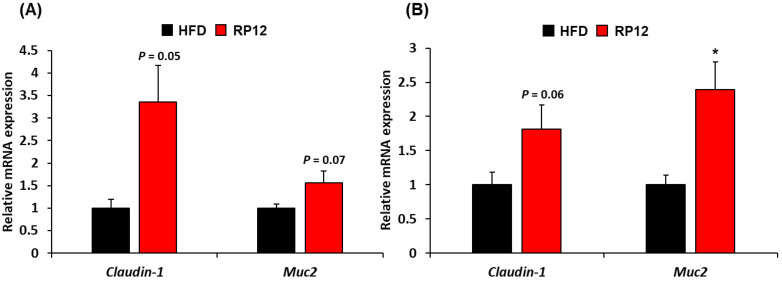
Effect of administered *Lactiplantibacillus plantarum* RP12 on expression of *Claudin-1* and *Muc2* in ileum and colon: (**A**) The mRNA expression levels of *Claudin-1* and *Muc2* in ileum measured by quantitative real-time PCR. (**B**) The mRNA expression levels of *Claudin-1* and *Muc2* in the colon measured using quantitative real-time PCR. Results are shown as mean ± SEM (*n* = 7). Significant differences between HFD and RP12 are indicated as * *p* < 0.05.

**Table 1 ijms-26-09056-t001:** Biochemical parameters of serum in the HFD and RP12 groups.

	HFD	RP12
Total cholesterol (mg/dL)	151.43 ± 10.51	139.40 ± 10.78 *
Glucose (mg/dL)	196.29 ± 15.60	125.80 ± 13.34 *
TG (mg/dL)	69.29 ± 6.19	61.40 ± 2.22
HDL (mg/dL)	95.43 ± 2.89	98.20 ± 2.22
LDL (mg/dL)	22.71 ± 1.81	18.80 ± 0.66

Values are shown as the means ± SEM (*n* = 7 per group). Abbreviation: TG, triglyceride; HDL, high-density lipoprotein; LDL, low-density lipoprotein. Significant differences between HFD and RP12 groups are indicated as * *p* < 0.05.

## Data Availability

The original contributions presented in this study are included in the article/Appendix A. Further inquiries can be directed to the corresponding author.

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
