# Peer review of "Anti-Obesity Effects and Changes of Fecal Microbiome by Lactic Acid Bacteria from Grains in a High-Fat Diet Mouse Model"

_ijms, 2025, doi:10.3390/ijms26189056_

Round 1
Reviewer 1 Report
Comments and Suggestions for Authors
Overall Assessment
This manuscript investigates the anti-obesity potential of Lactiplantibacillus plantarum RP12, isolated from grains, in a high-fat diet (HFD)-induced obese mouse model. The authors present body weight and adipose mass reduction, improvements in serum biochemical parameters, altered lipid metabolism gene expression, modulation of gut microbiota composition, increased fecal SCFAs, and preserved intestinal morphology with changes in barrier-related gene expression.
The topic is relevant to IJMS, and the scope fits well with the journal’s readership. The breadth of endpoints is commendable. However, the study has several weaknesses that limit the strength of its conclusions, including insufficient mechanistic validation, lack of quantitative histological analysis, overinterpretation of non-significant findings, inadequate statistical treatment (especially multiple comparisons), and incomplete methodological detail. A major revision is required to meet IJMS’s standards for scientific rigor, reproducibility, and data interpretation.
Major Comments
- Novelty and Framing
- The novelty of RP12 compared to other L. plantarum strains is not clearly articulated. Grain origin is emphasized, but the physiological or functional distinction from previously studied strains remains vague.
- The selection of RP12 for the second in vivo experiment (excluding K28 and RP21) is only briefly justified based on weight gain differences; more explicit rationale is needed.
- The term “therapeutic probiotic” is overstated for preclinical data without human evidence; “potential probiotic candidate” is more appropriate.
- Scientific Rigor and Statistical Analysis
- Sample size and statistical power: The main experiment uses n=7 per group, yet multiple endpoints (genes, taxa, metabolites) are tested without correction for multiple comparisons. False discovery rate (FDR) adjustment is necessary, particularly for gene expression and microbiome analyses.
- Beta diversity analysis: The PCoA separation is described visually, but no global statistical test (e.g., PERMANOVA with R² and p-value) is reported.
- LEfSe analysis: LEfSe results (LDA≥3, p<0.05) are presented without multiple comparison adjustment; this is prone to false positives. Authors should supplement with compositional data-aware methods (e.g., ANCOM-BC, ALDEx2) and report adjusted p-values.
- Data–Interpretation Consistency
- LDL cholesterol: Table 1 shows LDL reduction without statistical significance, yet the Discussion states that RP12 decreased LDL. This statement should be tempered or supported by additional data.
- Bacillota:Bacteroidota ratio: No difference is reported between groups, yet “gut microbiota modulation” is claimed broadly. Claims should be linked to lower taxonomic changes with demonstrated functional relevance.
- Fecal lipid and SCFA data: Only concentration (mg/g) is reported, not total excretion. Without total fecal output or energy balance data, conclusions about reduced lipid absorption are speculative.
- Barrier gene expression and histology: Claudin-1 and Muc2 mRNA increases are shown, but no protein-level confirmation (e.g., Western blot, IHC) or functional permeability assay is included. Linking mRNA changes to preserved barrier function requires additional validation.
- Mechanistic Depth
- Liver and adipose metabolism: Changes in gene expression (lipogenesis down, β-oxidation up) are reported without direct functional measures (enzyme activity, tissue lipid content beyond organ weight, metabolic data). These would strengthen mechanistic claims.
- Inflammatory gene caption: Figure 2B’s caption refers to “anti-inflammatory gene expression,” but the genes measured (Tnfα, Mcp1, Il-6) are pro-inflammatory cytokines whose expression is reduced. The caption and text should be corrected.
- Microbiome Analytics and Reproducibility
- No accession number for raw 16S rRNA data is provided; public deposition in SRA/ENA is required.
- Microbiome methods lack details on normalization, rarefaction depth, sequencing depth per sample, negative controls, and batch effect control. These should be specified for reproducibility.
- Taxonomic names are inconsistently spelled (e.g., Romboutsia appears as “Romboustia”). - Figure 7 Specific Concern
- Panels A and B (H&E images of ileum and colon) are purely qualitative with no quantitative histopathological metrics (e.g., villus height, crypt depth, goblet cell counts). Without such metrics, morphological preservation claims remain subjective.
- There is no direct linkage or correlation analysis between the histological images (A, B) and the mRNA expression data (C, D). Presenting them together implies a connection that is not statistically demonstrated.
- To strengthen this figure, authors should provide quantitative morphometry and, ideally, correlate these structural parameters with barrier gene expression.
Minor Comments
Language and Style
- Several typos (e.g., “fa y” instead of “fatty,” “a ributed” instead of “attributed”) and inconsistent italics for microbial names should be corrected.
- Some sentences are overly long and could be split for clarity.
Figures and Tables
- Each figure panel should include n-values, statistical test names, and significance thresholds in the captions.
- Replace “data not shown” with supplementary figures or tables where possible.
Methods
- SCFA analysis: internal standard, calibration curve details, LOD/LOQ should be described.
- qPCR: primer sequences should be listed in supplementary material rather than relying solely on references.
GTT data
- Provide individual mouse glucose curves and AUC calculations in supplementary data.
Ethics and Safety
- Specify whether the IACUC approval covered the exact study period.
- Provide safety assessment of RP12 (antibiotic resistance genes, mobile elements) or deposit strain in a public culture collection with accession number.
Language and Style
- Several typos (e.g., “fa y” instead of “fatty,” “a ributed” instead of “attributed”) and inconsistent italics for microbial names should be corrected.
- Some sentences are overly long and could be split for clarity.
Author Response
Reviewer 1
Major Comments
Novelty and Framing
- The novelty of RP12 compared to other L. plantarum strains is not clearly articulated. Grain origin is emphasized, but the physiological or functional distinction from previously studied strains remains vague.
Response: It was shown in our previous study (Seo et al. 2023) that strain RP12 is differentiated from some other L. plantarum strains by RAPD-PCR. Some physiological or functional properties of strain RP12 were provided in in our previous study (Seo et al. 2023). I think that the novelty or distinction may be not necessarily required for studies showing the anti-obesity effects of grain-derived lactic acid bacteria in mouse.
Seo, M.J.; Won, S.-M.; Kwon, M.J.; Song, J.H.; Lee, E.B.; Cho, J.H.; Park, K.W.; Yoon, J.-H. Screening of lactic acid bacteria with anti adipogenic efect and potential probiotic properties from grains. Sci. Rep. 2023, 13, 11022
- The selection of RP12 for the second in vivo experiment (excluding K28 and RP21) is only briefly justified based on weight gain differences; more explicit rationale is needed.
Response: For rationale, additional sentence “a more pronounced anti-obesity effect (showing 14-15% differences in weight gain) and significant improvement in glucose tolerance and AUC)” were inserted in line 287-288 in discussion section.
- The term “therapeutic probiotic” is overstated for preclinical data without human evidence; “potential probiotic candidate” is more appropriate.
Response: “therapeutic” was changed to “useful” in line 28. “therapeutic” was deleted in line 74.
Scientific Rigor and Statistical Analysis
- Sample size and statistical power: The main experiment uses n=7 per group, yet multiple endpoints (genes, taxa, metabolites) are tested without correction for multiple comparisons. False discovery rate (FDR) adjustment is necessary, particularly for gene expression and microbiome analyses.
Response: False discovery rate (FDR) adjustment was performed for microbiome analysis, which is major subject of this study as shown in the title.
- Beta diversity analysis: The PCoA separation is described visually, but no global statistical test (e.g., PERMANOVA with R² and p-value) is reported.
Response: The data were provided as suggested.
- LEfSe analysis: LEfSe results (LDA≥3, p<0.05) are presented without multiple comparison adjustment; this is prone to false positives. Authors should supplement with compositional data-aware methods (e.g., ANCOM-BC, ALDEx2) and report adjusted p-values.
Response: LEfSe analysis has been predominantly used in relevant papers when considering citation. Moreover, I think it is possible to show LEfSe results, because the number of samples is not large.
Data–Interpretation Consistency
- LDL cholesterol: Table 1 shows LDL reduction without statistical significance, yet the Discussion states that RP12 decreased LDL. This statement should be tempered or supported by additional data.
Response: “although these differences were not statistically significant” was given in line 127 of Result section.
- Bacillota:Bacteroidota ratio: No difference is reported between groups, yet “gut microbiota modulation” is claimed broadly. Claims should be linked to lower taxonomic changes with demonstrated functional relevance.
Response: In lines 368-369, 392-393 and 504, “the modulation of gut microbiota” was revised as “the modulation of several taxa in gut” to avoid the broad claim.
- Fecal lipid and SCFA data: Only concentration (mg/g) is reported, not total excretion. Without total fecal output or energy balance data, conclusions about reduced lipid absorption are speculative.
Response: I acknowledge the reviewer’s concern. In our study, the analysis was performed using the total feces collected from each mouse. Therefore, the reported concentrations reflect the complete fecal output. While we did not conduct a full energy balance analysis, the use of total feces allows us to more reliably interpret changes in fecal lipid and SCFA.
- Barrier gene expression and histology: Claudin-1 and Muc2 mRNA increases are shown, but no protein-level confirmation (e.g., Western blot, IHC) or functional permeability assay is included. Linking mRNA changes to preserved barrier function requires additional validation.
Response: I appreciate your insightful comment. Our intention in measuring Claudin-1 and Muc2 expression was to provide a preliminary survey of barrier-associated gene responses, rather than to comprehensively characterize intestinal permeability. Because the primary focus of this study was the anti-obesity effects of RP12, we did not pursue protein-level analyses or functional assays in this work. We agree that future studies specifically addressing epithelial barrier integrity will be required to establish a direct causal link between the observed gene expression changes and barrier function, and I have clarified this limitation in the revised Discussion.
Mechanistic Depth
- Liver and adipose metabolism: Changes in gene expression (lipogenesis down, β-oxidation up) are reported without direct functional measures (enzyme activity, tissue lipid content beyond organ weight, metabolic data). These would strengthen mechanistic claims.
Response: The scope of this study was limited to examining gene expression in mice that influence anti-obesity effects. It would be best if we could obtain the data you suggested, but it would likely be beyond the purpose of this study or the scope of the journal. Moreover, in many similar papers, it also appears to examine only the expression of those genes as auxiliary indicators of anti-obesity effects.
- Inflammatory gene caption: Figure 2B’s caption refers to “anti-inflammatory gene expression,” but the genes measured (Tnfα, Mcp1, Il-6) are pro-inflammatory cytokines whose expression is reduced. The caption and text should be corrected.
Response: The mistake was corrected.
Microbiome Analytics and Reproducibility
- No accession number for raw 16S rRNA data is provided; public deposition in SRA/ENA is required.
- Microbiome methods lack details on normalization, rarefaction depth, sequencing depth per sample, negative controls, and batch effect control. These should be specified for reproducibility.
Response: I thank the reviewer for this important point. The 16S rRNA sequencing and analysis were performed by ChunLab (currently CJ Bioscience, South Korea), a well-established provider of microbiome sequencing services. Unfortunately, due to the expiration of the company’s storage period, the raw sequencing data are no longer available for public deposition. Unfortunately, I did not take care of the raw 16S rRNA data for deposit, because our previous similar paper was published without deposit of the data in MDPI journal. I acknowledge that this is a limitation of the present study. However, all analyses were conducted using standardized pipelines provided by ChunLab, which are widely recognized and reproducible. I am sorry, but please understand the present situation not to deposit the data.
- Taxonomic names are inconsistently spelled (e.g., Romboutsia appears as “Romboustia”).
Response: It is confirmed that “Romboutsia” is correct. “Romboustia” was corrected as “Romboutsia”.
Figure 7 Specific Concern
- Panels A and B (H&E images of ileum and colon) are purely qualitative with no quantitative histopathological metrics (e.g., villus height, crypt depth, goblet cell counts). Without such metrics, morphological preservation claims remain subjective.
- There is no direct linkage or correlation analysis between the histological images (A, B) and the mRNA expression data (C, D). Presenting them together implies a connection that is not statistically demonstrated.
- To strengthen this figure, authors should provide quantitative morphometry and, ideally, correlate these structural parameters with barrier gene expression.
Response: I thank the reviewer for the valuable suggestion. I agree that the H&E images in Figures 7A and 7B provide only qualitative evidence. Similar qualitative histological comparisons have been presented in many previous microbiome and probiotic studies, and our initial intention was to illustrate representative morphological differences between the two groups. To address the reviewer’s concern, I revised the manuscript by moving these panels to the Supplementary Materials and by tempering the following description in the Discussion to avoid over-interpretation,
“even though no quantitative morphometric analysis (e.g., villus height and crypt depth) or correlation with gene expression was performed.
So please allow me the manuscript to modify by including them as supplementary materials.
Minor Comments
Language and Style
- Several typos (e.g., “fa y” instead of “fatty,” “a ributed” instead of “attributed”) and inconsistent italics for microbial names should be corrected.
Response: There appear no problems in MS word file.
- Some sentences are overly long and could be split for clarity.
Response: The manuscript was review again and modified properly.
Figures and Tables
- Each figure panel should include n-values, statistical test names, and significance thresholds in the captions.
Response: They were indicated in Material and Methods section. Only n-values were given in the captions to save space.
- Replace “data not shown” with supplementary figures or tables where possible.
Response: The manuscript was modified as suggested. “data not shown” was given as Supplementary figure.
Methods
- SCFA analysis: internal standard, calibration curve details, LOD/LOQ should be described.
Response: The details for the SCFA analysis have been given in reference cited.
- qPCR: primer sequences should be listed in supplementary material rather than relying solely on references.
Response: The primer sequences were provided as Supplementary Table.
GTT data
- Provide individual mouse glucose curves and AUC calculations in supplementary data.
Response: The data provided as supplementary figure.
Ethics and Safety
- Specify whether the IACUC approval covered the exact study period.
Response: It was specified in Materials and Methods section.
- Provide safety assessment of RP12 (antibiotic resistance genes, mobile elements) or deposit strain in a public culture collection with accession number.
Response: The safety assessment may be necessary for administration to human, but it may not be necessarily required at this time. Instead, it was confirmed that the strain RP12 is unlikely to cause safety concerns due to inherent resistance from sensibility data to antibiotics of strain RP12 provided in our previous study (Seo et al., 2023). I think that it may not be necessarily required to deposit the strain in a public culture collection for publication in IJMS, because it may be registered as a patented strain.
Reviewer 2 Report
Comments and Suggestions for Authors
- It is strongly recommended to include the assessment of both alpha diversity and beta diversity to present the structure and changes of the microbial community more clearly.
- To more accurately assess the intestinal morphology and barrier function, the authors should consider measuring the intestinal villus height (IVH), crypt depth (ICD), and the ratio of villi to crypts (H/D). These parameters are widely used as indicators of intestinal barrier integrity. For methodological references, please refer to: https://doi.org/10.3390/biology140504963.
- The method for determining the sample size is not specified. Please provide detailed information on how the number of animals was selected and whether it was based on power analysis or previous studies as a basis.
- The visualization method of LEfSe results is not uniform. It is recommended to use the conventional LEfSe output format to redraw these charts.
- Since Claudin-1 and Muc2 are important components of the intestinal barrier, if these conclusions can be confirmed through immunohistochemical staining, their persuasiveness will be stronger. Additionally, including other tight junction markers, such as ZO-1 and occludin, will enable a more comprehensive assessment of the barrier integrity.
- Please discuss the advantage and limitation of Lactiplantibacil lusplantarum RP12 compared to other method or stains.
Author Response
Reviewer 2
- It is strongly recommended to include the assessment of both alpha diversity and beta diversity to present the structure and changes of the microbial community more clearly.
Response: The data have been provided as Supplementary Figure 4.
- To more accurately assess the intestinal morphology and barrier function, the authors should consider measuring the intestinal villus height (IVH), crypt depth (ICD), and the ratio of villi to crypts (H/D). These parameters are widely used as indicators of intestinal barrier integrity. For methodological references, please refer to: https://doi.org/10.3390/biology140504963.
Response: Response: I appreciate your insightful comment. Our intention in measuring Claudin-1 and Muc2 expression was to provide a preliminary survey of barrier-associated gene responses, rather than to comprehensively characterize intestinal permeability. Because the primary focus of this study was the anti-obesity effects of RP12, we did not pursue protein-level analyses or functional assays in this work. We agree that future studies specifically addressing epithelial barrier integrity will be required to establish a direct causal link between the observed gene expression changes and barrier function, and I have clarified this limitation in the revised Discussion. These qualitative data have been also used in similar many papers, so please allow me the manuscript to modify by including them as supplementary materials.
- The method for determining the sample size is not specified. Please provide detailed information on how the number of animals was selected and whether it was based on power analysis or previous studies as a basis.
Response: The information was provided in 4.2. section.
- The visualization method of LEfSe results is not uniform. It is recommended to use the conventional LEfSe output format to redraw these charts.
Response: LEfSe results were modified according to the comment. I wonder whether the modification is proper.
- Since Claudin-1 and Muc2 are important components of the intestinal barrier, if these conclusions can be confirmed through immunohistochemical staining, their persuasiveness will be stronger. Additionally, including other tight junction markers, such as ZO-1 and occludin, will enable a more comprehensive assessment of the barrier integrity.
Response: My apology to similar comments is also shown above. The experiment was just to survey the effect on permeability through the expression of the two genes, because the subjects of this study focus on anti-obesity, not about permeability or junction proteins.
- Please discuss the advantage and limitation of Lactiplantibacil lusplantarum RP12 compared to other method or stains.
Response: I'm not sure if it's appropriate, but the following contents have been newly added to the discussion section. Lactiplantibacillus plantarum RP12, which was isolated from grains, may be particularly advantageous at utilizing the carbohydrates present as prebiotics in the gut. Nevertheless, its effectiveness appears to have limitation that requires tests on humans or animals in the future.
Round 2
Reviewer 1 Report
Comments and Suggestions for Authors The revised paper has been revised to reflect the reviewer's comments and has been confirmed.Reviewer 2 Report
Comments and Suggestions for Authors
Accept in present form